# Polylogarithmic-depth controlled-NOT gates without ancilla qubits

Baptiste Claudon [1,2] ✉, Julien Zylberman [3], César Feniou[1,2], Fabrice Debbasch[3], Alberto Peruzzo[1] & Jean-Philip Piquemal [1,2] ✉

Controlled operations are fundamental building blocks of quantum algorithms. Decomposing $n$-control-NOT gates ($C^n(X)$) into arbitrary single-qubit and CNOT gates, is a crucial but non-trivial task. This study introduces $C^n(X)$ circuits outperforming previous methods in the asymptotic and non-asymptotic regimes. Three distinct decompositions are presented: an exact one using one borrowed ancilla with a circuit depth $\Theta(\log(n)^3)$, an approximating one without ancilla qubits with a circuit depth $\mathcal{O}(\log(n)^3 \log(1/\epsilon))$ and an exact one with an adjustable-depth circuit which decreases with the number $m \leq n$ of ancilla qubits available as $\mathcal{O}(\log(n/\lfloor m/2 \rfloor)^3 + \log(\lfloor m/2 \rfloor))$. The resulting exponential speedup is likely to have a substantial impact on fault-tolerant quantum computing by improving the complexities of countless quantum algorithms with applications ranging from quantum chemistry to physics, finance and quantum machine learning.

In the past three decades, quantum algorithms promising an exponential speedup over their classical counterparts have been designed[1-3]. The advantages of these algorithms stem from the peculiar properties of superposition and entanglement of the quantum bits (qubits). These properties enable the manipulation of a vast vector of qubit states using basic operations that act on either one or two qubits[3]. The question of decomposing efficiently any $n$-qubit operation into a reasonable number of primitive single- and two-qubit operations is one of the major challenges in quantum computing[4]. In this context, multi-controlled operations act as building blocks of many prevalent quantum algorithms such as qubitisation[5] within the quantum singular value transformation[6], which have immediate repercussions on Hamiltonian simulation[7], quantum search[6] and quantum phase estimation methods[8]. For this reason, achieving more effective decompositions of multi-controlled operations has the potential to bring about significant enhancements in quantum algorithms, impacting various fields such as quantum chemistry, specifically in estimating ground state energy[9], physics for the simulation of quantum systems[10], engineering for solving partial differential equations[11,12], quantum machine learning[13], and finance[14].

This non-trivial challenge and the quest for optimal solutions have been an active and ongoing research focus for decades. In 1995, Barenco et al.[15] proposed several linear depth constructions of multi-controlled NOT (MCX) gates (also known as $n$-Toffoli gates, multi-controlled Toffoli, generalised-Toffoli gates, multi-controlled $X$ or $n$-controlled NOT gates). All these linear depth constructions used ancilla qubits or relied on an efficient approximation, while the first ancilla-free exact decomposition has a quadratic depth.

Years later, exact implementations of multi-controlled NOT gates with linear depth and quadratic size were proposed[16,17]. It was only in 2015 that Craig Gidney published a pedagogical blog post describing an exact linear size decomposition without ancilla qubits[18]. Even though his method was linear, its depth leading coefficient is large compared to that of other methods. Still, this one-of-a-kind method was used in subsequent work[19-21]. In 2017, a logarithmic-depth multi-controlled NOT gate using as many zeroed ancilla qubits as control qubits was presented[22]. Such a finding motivated the search for a trade-off between the number of ancillae and the circuit depth. Computational approaches have been implemented[23], suggesting that any zeroed ancilla qubit could be used to reduce the circuit depth of

[1]Qubit Pharmaceuticals, Advanced Research Department, Paris, France. [2]Sorbonne Université, LCT, UMR 7616 CNRS, Paris, France. [3]Sorbonne Université, Observatoire de Paris, Université PSL, CNRS, LERMA, Paris, France. ✉e-mail: baptiste.claudon@qubit-pharmaceuticals.com; jean-philip.piquemal@sorbonne-universite.fr

controlled operations. Recently, Orts et al.[24] provided a review of the 2022 state-of-the-art methods for MCX.

Every multi-controlled NOT decomposition method aims at optimising certain metrics such as circuit size, circuit depth, or ancilla count. This article considers only the case of $n$-controlled NOT gates with $n > 2$ (he particular problem of decomposing the 2-controlled NOT gate, i.e., the Toffoli gate, possesses distinct optimal solutions for various metrics concerning both single- and two-qubit gates[25,26]). The attention is directed towards circuit depth, the metric associated with quantum algorithmic runtime, circuit size, the total number of primitive quantum gates (computed on the basis of arbitrary single-qubit and CNOT gates), and ancilla qubit count, which corresponds to the amount of available qubits during the computation. A precise distinction is made between borrowed ancilla qubits (qubits in any state beforehand which are restored afterwards) and zeroed ancilla qubits (which are in state $|0\rangle$ initially and reset to $|0\rangle$ at the end of the computation). Zeroed ancilla qubits are efficient to use since they are initially in a well-known state. Borrowed ancilla qubits are more constraining but have the advantage of being available more often during computations: operations that do not impact the entire system can utilise unaffected wires as borrowed qubits.

This article introduces three circuit identities to build a $n$-controlled NOT gate $C^n(X)$. They display the following depth complexities.

1. Polylogarithmic-depth circuit with a single borrowed ancilla (Proposition 1).
2. Polylogarithmic-depth approximate circuit without ancilla (Proposition 2).
3. Adjustable depth circuit using an arbitrary number $m \leq n$ of ancillae (Proposition 3).

The first one is an exact decomposition whose depth is affine in the depth of a quadratically smaller controlled operation $C^{\sqrt{n}}(X)$. The global decomposition takes advantage of a single borrowed ancilla qubit, and each of the smaller controlled operations can be associated with a locally borrowed ancilla. As a consequence, a recursive construction of the multi-controlled NOT gate emerges with an overall polylogarithmic circuit depth in the number of control qubits $\Theta(\log(n)^3)$ and a circuit size $\mathcal{O}(n\log(n)^4)$. The second method approximates a $C^n(X)$ without ancilla qubits up to an error $\epsilon > 0$ with a circuit depth $\mathcal{O}(\log(n)^3\log(1/\epsilon))$ and a circuit size $\mathcal{O}(n\log(n)^4\log(1/\epsilon))$. It makes use of the decomposition of a $n$-controlled $X$ gate into $(n-1)$-controlled unitaries, generating borrowed ancilla qubits that facilitate the application of the first method up to an error $\epsilon$. The last method provides an adjustable-depth quantum circuit which implements exactly a $C^n(X)$ for any given number $m \leq n$ of zeroed ancilla qubits. The depth decreases with $m$ from a polylogarithmic-with-$n$ scaling to a logarithmic one as $\mathcal{O}(\log(n/\lfloor m/2 \rfloor)^3 + \log(\lfloor m/2 \rfloor))$. To the best of our knowledge, these methods stand out as the only approaches achieving such depth complexities. In particular, they demonstrate an exponential speedup over previous state-of-the-art (or, more formally, a superpolynomial speedup[27]) and readily improve a wide range of quantum algorithms.

## Results

This section is subdivided into three parts. The first subsection details the logical steps towards achieving method 1. This includes a detailed study of the single-zeroed-ancilla case, how it can be turned into a borrowed one and where the recursive decomposition can stem from to yield the polylogarithmic-depth controlled-NOT circuit with a single borrowed ancilla. The methods 2 and 3 are described respectively in IIB and IIC.

### Notations

This paragraph gathers the most important definitions that will be used throughout the article. Let $n \geq 1$ and let $\{|x\rangle\}_{x=0}^{2^n-1}$ be the computational basis of an $n$-qubit register $R$ and $T$ be a one-qubit register, the

so-called target register. The $C^n(X)$ controlled by $R$ with target $T$ is the gate defined by equation (1):

$$C^n(X) = (I_R - |2^n-1\rangle\langle 2^n-1|) \otimes I_T \\ + |2^n-1\rangle\langle 2^n-1| \otimes X, \quad (1)$$

where $I_R$ and $I_T$ are the identities on register $R$ and $T$. The symbol $X$ denotes the Pauli matrix $X = |0\rangle\langle 1| + |1\rangle\langle 0|$. More generally, a $C^n(U)$ gate will denote a controlled $U$ gate. If the register $R$ is in state $|2^n-1\rangle$, it will be termed as active. When one wants to emphasise the control and target registers, one will note $\mathcal{C}_R^T$. If $R'$ is a second qubit register, define the (partial) white control $\mathcal{C}_{\overline{R} \cup R'}^T \equiv (\prod_{q\in R} X_q)\mathcal{C}_{R\cup R'}^T(\prod_{q\in R} X_q)$, where $X_q$ denotes an $X$ gate on white control qubit $q$. The support of a quantum circuit is the set of qubits on which it does not act like the identity. Ancilla qubits are qubits outside the support of $U$ which may be used during the computation. From now, the function log signifies the logarithm in base 2. All circuit sizes and depths are computed on the basis of arbitrary single-qubit and CNOT gates.

### Polylogarithmic-depth gate using one borrowed ancilla

The decomposition uses a single zeroed ancilla to reduce the global MCX gate to five layers, each exhibiting the depth of a quadratically smaller operation. The $n$-control-qubit register $R = \{q_0, ..., q_{n-1}\}$ is first divided into subregisters. Let $p = \lfloor\sqrt{n}\rfloor$. The control register $R$ can then be written as the disjoint union $R = \bigcup_{i=0}^b R_i$ where subregister $R_0 = \{q_0, ..., q_{2p-1}\}$ is a subregister of size $2p$ and subregister $R_i$ is of size at most $p$, for each $i \in \{1, ..., p\}$. More precisely, let $R_i = \{q_j\}_{j=(1+i)p}^{(2+i)p-1}$ for each $i \in \{1, ..., p\}$. Let $b = |\{i \in \{1, ..., p\} : R_i \neq \emptyset\}|$ be the number of non-empty registers of positive index. Moreover, the first register $R_0$ is further divided into $R_0^* = \{q_i\}_{i=0}^{b-1}$ containing the first $b$ qubits and $R_0' = R_0 \backslash R_0^*$.

The circuit in Fig. 1 illustrates a decomposition of the controlled-NOT operation $\mathcal{C}_R^t$ using a single zeroed ancilla. Note that qubit $q_i$ is positioned between registers $R_i$ and $R_{i+1}$. The unitary associated with the circuit is given by

$$\mathcal{U}_0 \equiv \mathcal{C}_{R_0}^a \mathfrak{C} \mathcal{C}_{R_0}^a, \quad (2)$$

where

$$\mathfrak{C} \equiv \left(\prod_{i=1}^b \mathcal{C}_{R_i}^{q_{i-1}}\right) \mathcal{C}_{\overline{R_0^*} \cup a}^t \left(\prod_{i=1}^b \mathcal{C}_{R_i}^{q_{i-1}}\right). \quad (3)$$

The first operation $\mathcal{C}_{R_0}^a$ sets the zeroed ancilla to $|1\rangle$ if and only if $R_0$ is active. Now, consider the operation $\mathfrak{C}$: for $\mathcal{C}_{\overline{R_0^*} \cup a}^t$ to apply an $X$ gate to the target, all the qubits from $R_0^*$ must be in state $|0\rangle$, and the ancilla $a$ must be in state $|1\rangle$. Now, notice that for each qubit $q_i \in R_0^*$, $q_i$ is in state $|0\rangle$ when applying $C_{\overline{R_0^*} \cup a}^t$ if and only if it was in the same activation state as $R_{i+1}$ right before applying $\mathfrak{C}$. Therefore, $\mathfrak{C}$ is a multi-controlled-$X$ gate conditioned on:

- the ancilla $a$ being in state $|1\rangle$ and
- for each index $i \in \{0, ..., b-1\}$: $q_i$ being in the same activation state as $R_{i+1}$.

By inspection, in the circuit in Fig. 1, these conditions are fulfilled if and only if the register $R$ of interest is active. Therefore, the target $t$ is flipped under exactly the right hypothesis. The operations following the potential flip of qubit $t$ leave it unchanged and reset both the qubits from $R_0^*$ and the ancilla to their initial state. The overall operation is the desired $\mathcal{C}_R^t$ gate. A more formal proof is given in Supplementary Note 1.

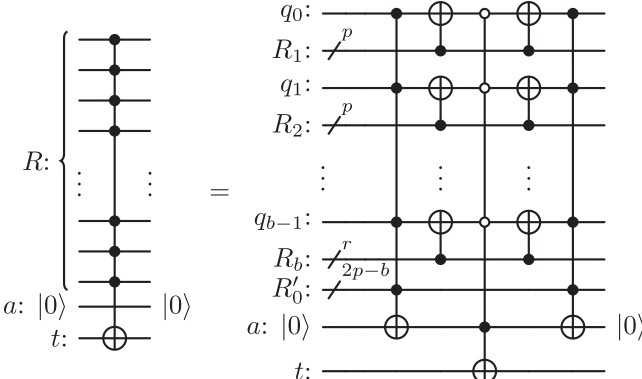

**Fig. 1 | $n$ controlled $\mathcal{C}_R^t$ using the zeroed ancilla $a$, where $p = \lfloor\sqrt{n}\rfloor$ and $R_i$ is a register of at most $p$ qubits for each $i \in \{1, ..., b\}$.** $r$ is the remainder of the euclidean division of $|R\backslash R_0| = n - 2p$ by $p$, $r = |R_b|$.

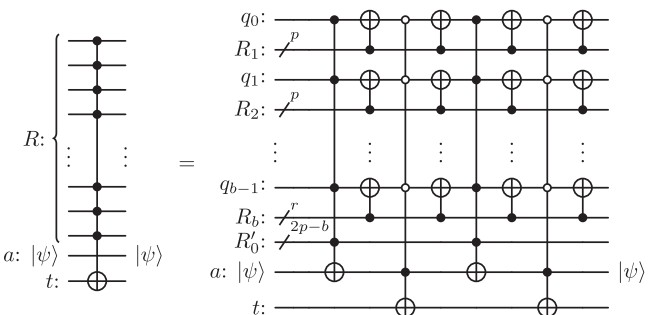

**Fig. 2 | $n$ controlled $\mathcal{C}_R^t$ using the borrowed ancilla $a$, where $p = \lfloor\sqrt{n}\rfloor$ and $R_i$ is a register of at most $p$ qubits for each $i \in \{1, ..., b\}$.** $r$ is the remainder of the euclidean division of $|R\backslash R_0| = n - 2p$ by $p$, $r = |R_b|$.

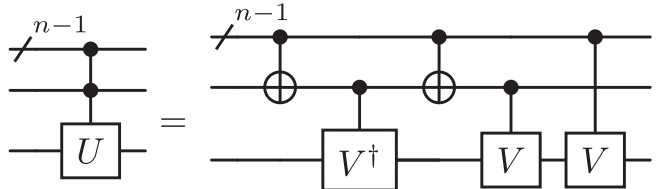

**Fig. 3 | Controlling arbitrary unitary operation with a zeroed ancilla qubit.** $U$ is the unitary to control, $V$ is a square root of $U$ and $n$ is the number of control qubits.

This subsection discusses how the above zeroed ancilla is transformed into a borrowed ancilla. The construction is displayed in the circuit in Fig. 2, which is formally defined by the unitary operation

$$\mathcal{U} \equiv \mathfrak{C}\, \mathcal{C}_{R_0}^a\, \mathfrak{C}\, \mathcal{C}_{R_0}^a = \mathfrak{C}\, \mathcal{U}_0. \tag{4}$$

Notice that $\mathcal{U}_0$ and $\mathfrak{C}$ both leave the control qubits and the ancilla unchanged. As a consequence, one only needs to focus on what happens to the target $t$. If the ancilla starts in state $|0\rangle$, the last $\mathfrak{C}$ does not affect $t$ and $\mathcal{U}$ overall acts as the desired $\mathcal{C}_R^t$ gate. It is now sufficient to focus on the case where qubit $a$ arrives in state $|1\rangle$ and to repeat the above analysis: $\mathcal{U}_0$ flips the target if both $R_0$ is inactive and for each $i \in \{0, ..., b-1\}$, qubit $q_i$ is in the same activation state as $R_{i+1}$. Refer to this flip as the first flip. Then, $\mathfrak{C}$ flips the target if and only if for each $i \in \{0, ..., b-1\}$, qubit $q_i$ is in the same activation state as $R_{i+1}$. Refer to this flip as the second flip. Summarising, if $R$ is active only the second flip occurs. If $R$ is inactive, there are two cases: $R_0$ is either inactive or active. If $R_0$ is inactive, both flip 1 and flip 2 occur, resulting in no flip at all. If $R_0$ is active, the second condition cannot be verified. Therefore,

none of the two flips occurs. In any case, the circuit applies the $\mathcal{C}_R^t$ gate and leaves the ancilla unaffected. A more thorough proof is given in Supplementary Note 1.

It is now possible to use a borrowed ancilla to implement any $C^n(X)$ using smaller $C^{2p}(X)$, $C^{p+1}$ and $C^p(X)$ gates. This section details how a qubit can be borrowed to compute each of the smaller MCX operations. This allows the setting up of a recursive construction, keeping the number of ancillae to one and achieving the polylogarithmic depth decomposition of n-MCX.

In Fig. 2, the block $\prod_{i=1}^{b} \mathcal{C}_{R_i}^{q_{i-1}}$ has the same depth as a single controlled-NOT gate with $p$ control qubits since it is the product of $b$ unitaries with disjoint support. The depth of a controlled operation containing white controls is the same as that of standard controlled-NOT plus that of two additional layers of X gates acting on qubits whose control colour is white. One may want to apply the decomposition from the circuit in Fig. 2 to each $C^p(X)$ gate. In order to do so, each block must have a borrowed ancilla qubit at its disposal.

There are more available borrowed qubits in the register $R_0'$ than operations to perform in parallel: $|R_0'| \geq |R_0^*|$ (see Supplementary Note 1). Hence, there are sufficiently many borrowed ancilla qubits to construct the smaller operations recursively. The recursion gives the following equality for the depth $D_n$ of a $C^n(X)$ gate.

$$D_n = 2D_{2p} + 4D_p + 2D_{b+1} + 4, \tag{5}$$

with $p = \lfloor\sqrt{n}\rfloor$ and $p - 2 \leq b \leq p$. The asymptotic behaviour of the depth can be studied as follow: let $(\mathcal{D}(k))_{k\in\mathbb{N}} \equiv (D_{2^{k+2}})_{k\in\mathbb{N}}$ be the circuit depth of $C^{2^{k+2}}(X)$ gates. Equation (5) implies:

$$\mathcal{D}(k) \leq 8\mathcal{D}(k/2) + 4. \tag{6}$$

The Master Theorem, recalled in Supplementary Note 2, implies that $\mathcal{D}(k) \in \mathcal{O}(k^3)$. In terms of $n$, the circuit depth of a $C^n(X)$ gate using one borrowed ancilla qubit is $D_n \in \mathcal{O}(\log(n)^3)$. Now, a lower bound is derived by defining $(\tilde{\mathcal{D}}(k))_{k\in\mathbb{N}} \equiv (D_{2^k})_{k\in\mathbb{N}}$ such that $\tilde{\mathcal{D}}(k) \geq 8\tilde{\mathcal{D}}(k/2) + 4$. Similarly, this inequality leads to $D_n \in \Omega(\log(n)^3)$. The scaling of the size is computed similarly to the scaling of the depth, i.e., by solving the corresponding recursive equation. Details are given in Supplementary Note 3. The following proposition gathers these statements.

**Proposition 1.** A controlled-NOT gate with $n$ control qubits is implementable with a circuit of depth $\Theta(\log(n)^3)$, size $\mathcal{O}(n\log(n)^4)$ and using a single borrowed ancilla qubit through the recursive use of the circuit in Fig. 2.

A first application improved by Proposition 1 affects the decomposition of multi-controlled unitary $C^n(U)$ using one zeroed ancilla. The zeroed ancilla allows the decomposed $C^n(U)$ gate into two $C^n(X)$ gates and one $C^1(U)$ using the circuit identity in Fig. 3 (Lemma 7.11 from[15]).

The following corollary gives the complexity to implement a $C^n(U)$ gate.

**Corollary 1.** Let $U$ be a unitary of size $S$ in the basis of single-qubit gates and CNOT gates. A controlled $U$ gate with $n$ control qubits is implementable with depth $\mathcal{O}(S + \log(n)^3)$, size $\mathcal{O}(S + n\log(n)^4)$ and one zeroed ancilla qubit through the circuit in Fig. 3.

It is also straightforward to implement any special unitary single-qubit gate $W \in SU(2)$ with polylogarithmic complexity and without ancilla. This can be done with the help of Lemma 7.9 in ref. 15, illustrated in Fig. 4.

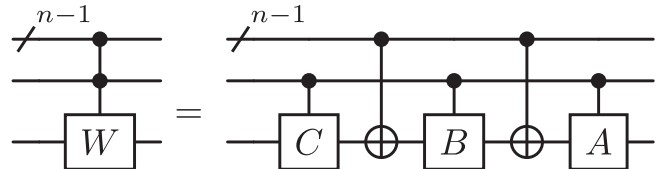

**Fig. 4 | Controlling arbitrary special unitary operation W.** Without loss of generality, $W = AXBXC$ for some matrices $A, B, C \in \mathrm{SU}(2)$ such that $ABC = I$.

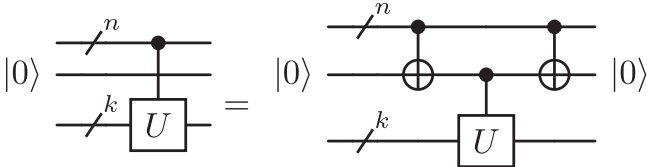

**Fig. 5 | Decomposition of a controlled unitary operation into controlled operations with fewer control qubits.** $U$ is the unitary to control, $n$ is the number of control qubits and $k$ is the number of qubits on which the gate $U$ acts.

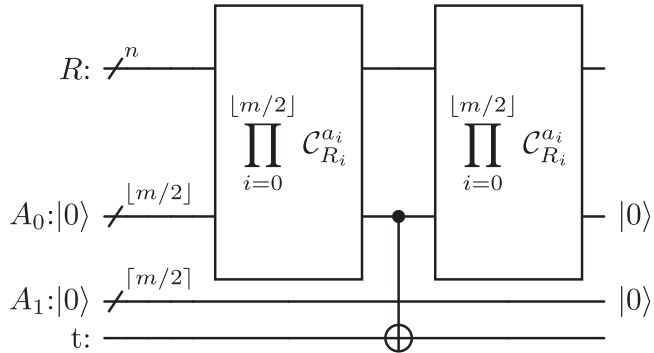

**Fig. 6 | Decomposition of a controlled-NOT gate into controlled operations with fewer control qubits using zeroed ancilla qubits.** $R$ is the control register of size $n$, $R_i$ are subregisters of $R$. $A_i$ are registers of zeroed ancilla qubits containing the qubits $a_i$, $i \in \{0, \dots, m-1\}$. $t$ denotes the target qubit.

**Corollary 2.** Let $W$ be a special unitary single-qubit gate. A $C^n(W)$ operation can be implemented with a circuit of depth $\mathcal{O}(\log(n)^3)$ and size $\mathcal{O}(n \log(n)^4)$ without ancilla qubits.

### Polylogarithmic-depth and ancilla-free approximate gate

This section outlines how to make use of Proposition 1 to control single-qubit unitaries in the absence of ancilla qubits. The first step generates borrowed ancilla qubits by decomposing a $n$-controlled unitary into $(n-1)$-controlled unitaries. More precisely, for any single-qubit unitary $U$ whose square root is denoted by $V$, one can implement a $C^n(U)$ gate from two $C^{n-1}(X)$, a $C^{n-1}(V)$ and two simple two-qubit gates using the circuit identity in Fig. 5 (Lemma 7.8 in ref. 15).

Applying this decomposition $k$ times involves implementing the $2^k$-th root $V_k$ of the original unitary. Performing this recursion $n$ times leads to a linear depth quantum circuit[18]. To circumvent this issue, it is possible to neglect the $(n-k)$-controlled $V_k$ gate, introducing an error exponentially small with $k$ (Lemma 7.8 in ref. 15):

$$\left\| C^{n-k}(V_k) - C^{n-k}(I) \right\| \leq \pi/2^k. \tag{7}$$

Applying the recursion $k = \lceil \log(\pi/\epsilon) \rceil$ times gives an $\epsilon > 0$ error on the implementation of the $C^n(U)$ gates. This decomposition leads to $2k$ one-controlled-root of $U$ and $2k$ multi-controlled NOT gates. Each MCX gate is controlled by a number $j \in \{1, \dots, n-1\}$ of qubits and, therefore,

is implementable using one of the non-affected qubits as a borrowed ancilla through the first method 1 with a polylogarithmic depth. The following proposition summarises the complexity of the method.

**Proposition 2.** For any single-qubit unitary $U \in \mathrm{U}(2)$, a controlled-$U$ gate with $n$ control qubits is implementable up to an error $\epsilon > 0$ (in spectral norm) with a circuit of depth $\mathcal{O}(\log(n)^3 \log(1/\epsilon))$, size $\mathcal{O}(n \log(n)^4 \log(1/\epsilon))$ without ancilla qubits.

The size and depth of the quantum circuits are trivially bounded noticing that the computational cost of $kj$-MCX, $j \leq n-1$, is bounded by the computational cost of $kn$-multi-controlled NOT gates.

This approximation proves highly effective for practical applications, as it is not needed to implement gates with exponentially small phases. The error $\epsilon > 0$ can be selected to align with the intrinsic hardware error, providing a level of flexibility that exact methods approaches may not offer.

### Adjustable-depth method

This subsection explains how to implement a $C^n(X)$ gate given $2 \leq m \leq n$ zeroed ancilla qubits. For simplicity, consider an even number of ancillae $m$ and a number of control qubits $n$ divisible by $m/2$. The method has three steps: a first one where $m/2$ controlled-NOT gates, each controlled by $n/(m/2)$ qubits, are performed in parallel in order to store the activation of the sub-registers into $m/2$ ancillae. A second step where a $C^{m/2}(X)$ controlled by the first $m/2$ ancillae is implemented on the target qubit using the last $m/2$ zeroed ancillae. A last one to restore the ancilla qubits in state $|0\rangle$. The first step makes use of Proposition 1 to implement each $C^{n/(m/2)}(X)$ with depth $\Theta(\log(n/(m/2))^3)$ using one zeroed ancilla of the last $m/2$ ancillae. The second step uses the logarithmic method from[22] to implement the $C^{m/2}(X)$ with depth $\mathcal{O}(\log(m/2))$ using $m/2$ ancillae. The circuit in Fig. 6 represents the three steps of the adjustable-depth method.

More generally, one can consider any value of $n$ control qubits and $m$ ancilla qubits such that $2 \leq m \leq n$ qubits. Let $A = \{a_0, \dots, a_{m-1}\}$ be the register of zeroed ancilla qubits, $A_0 = \{a_0, \dots, a_{\lfloor m/2 \rfloor - 1}\}$ and $A_1 = \{a_{\lfloor m/2 \rfloor}, \dots, a_{m-1}\}$ such that $A = A_0 \cup A_1$ and let the control register $R$ be divided into $\lfloor m/2 \rfloor$ balanced subregisters $(R_i)_{i=0}^{\lfloor m/2 \rfloor - 1}$. Let $\Pi$ be the operation associated to the first step:

$$\Pi \equiv \prod_{i=0}^{\lfloor m/2 \rfloor - 1} \mathcal{C}_{R_i}^{a_i}. \tag{8}$$

Each of the $\mathcal{C}_{R_i}^{a_i}$ is implemented in parallel of the others using the circuit in Fig. 1 with one zeroed ancilla of register $A_1$. Since all the $\mathcal{C}_{R_i}^{a_i}$'s are performed in parallel, the operation $\Pi$ has the same depth as the maximum depth of the $\mathcal{C}_{R_i}^{a_i}$'s. After applying $\Pi$, one can implement $\mathcal{C}_{A_0}^{t}$ using the method from[22], which uses as many ancillae as control qubits to achieve a logarithmic depth. Therefore, one can implement $\mathcal{C}_{A_0}^{t}$ with depth $16 \lceil \log(\lfloor m/2 \rfloor) \rceil + 12$ using $A_1$ as a register of zeroed ancilla qubits (Corollary 2 in[22]). Finally, one can repeat $\Pi$ to fully reset the register $A$. Proposition 3 summarises the complexity of this new method.

**Proposition 3.** Let $n \geq 2$ be the number of control qubits and $2 \leq m \leq n$ be the number of available zeroed ancillae. Then, there exists a decomposition of the $C^n(X)$ into single-qubit gates and CNOT gates with depth complexity:

$$\mathcal{O}\left( \log(n/\lfloor m/2 \rfloor)^3 + \log(\lfloor m/2 \rfloor) \right). \tag{9}$$

Note that the depth decreases as the number of ancillae $m$ increases, providing a method with adjustable depth. This is a valuable asset for aligning with the constraints of the hardware resources. Also, remark that the method from[22] has been used only for the second part of the algorithm. Using it in the first part would require a number of

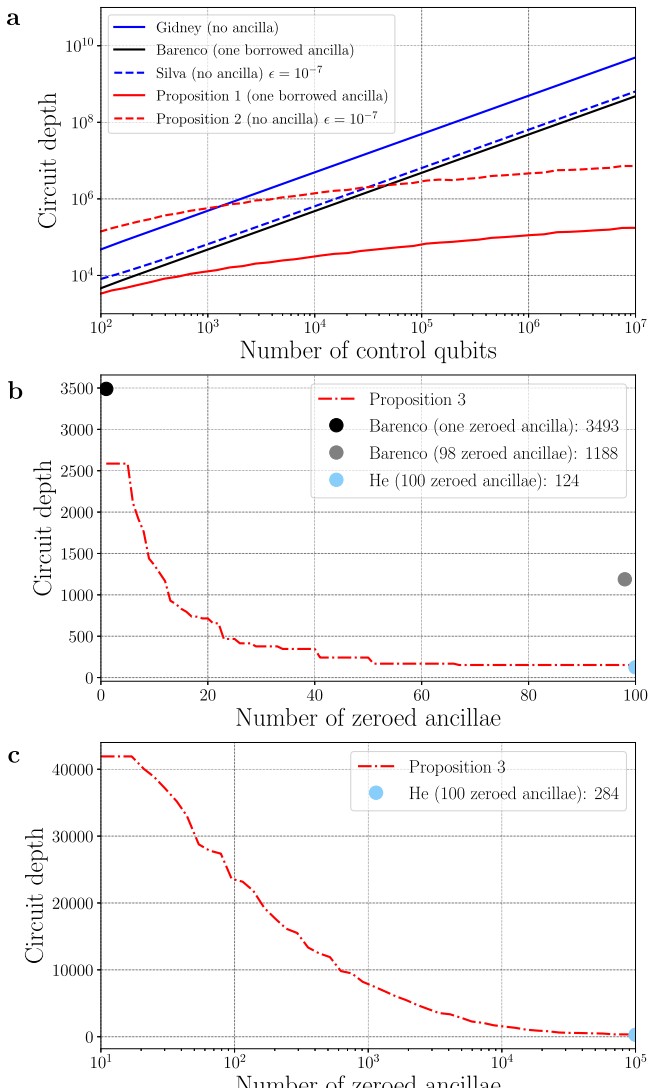

**Fig. 7 | Numerical analysis of circuit depths. a** Comparison of circuit depths for methods involving a single or no ancilla. **b** $C^{100}(X)$ depth as a function of the number of zeroed ancillae. **c** $C^{10^5}(X)$ depth as a function of the number of zeroed ancillae.

ancilla qubits proportional to the number of blocks, thus be potentially large. Different combinations of methods did not seem to lead to particular improvements in terms of depth or size. Overall, this decomposition provides a range of logarithmic-depth methods using less than $n$ ancillae, by considering $m(n) = \alpha n$, with $0 < \alpha < 1$, improving the decomposition proposed in[22].

## Discussion

In non-asymptotic regimes, pre-factors play a crucial role. This section employs graphical comparisons and complexity tables to fairly evaluate the overall effectiveness of current MCX decomposition methods. The circuits are compiled on the basis of single-qubit and CNOT gates for the number of control qubits ranging from $10^2$ to $10^7$. The obtained depths are numerically fitted, giving useful estimates for each method. Since this section only aims to serve as a resource estimator, the linear depth decompositions are fitted with a first-order polynomial in the variable $n$ and the depth of the decomposition from Proposition 1 is fitted with a first-order polynomial in the variable $\log(n)^3$.

In the case where only one zeroed borrowed ancilla qubit is available, a comparison is made with the one ancilla method from[15] as well as Craig Gidney's ancilla-free method, and illustrated in Fig. 7a. In practice, the recurrence from Proposition 1 must be initialised. For a

## Table 1 | Circuit depth of ancilla-free methods

| Method | Circuit depth |
|---|---|
| Gidney[18] | $494n - 1413$ |
| Silva ($\epsilon$)[28] | $64n + 1645$ if ($\epsilon = 10^{-7}$) |
| Proposition 2 ($\epsilon$) | $\lceil \log(\pi/\epsilon) \rceil (86 \log(n)^3 - 2564)$ |

The depths are numerically fitted in the range from $10^2$ to $10^7$ control qubits.

## Table 2 | Circuit depth of borrowed ancillae methods

| Method | Circuit depth | Borrowed |
|---|---|---|
| Barenco 1[15] | $48n - 148$ | 1 |
| Barenco $n-2$[15] | $24n - 43$ | $n - 2$ |
| Proposition 1 | $43 \log(n)^3 - 1287$ | 1 |

The depths are numerically fitted in the range from $10^2$ to $10^7$ control qubits.

number of control qubits $n$ less than 30, the $C^n(X)$ gate is implemented using Barenco et al.'s single-borrowed-ancilla method (the small gates can also be optimised via a brute-force approach). From there, the circuits and depths can be computed with a simple dynamic programming approach. This implementation provides a shallower circuit for any number of control qubits. In that case, the asymptotic advantage of Proposition 1 becomes evident early in the process, making the method applicable across various regimes.

Next, in the absence of an ancilla qubit but allowing an approximation error $\epsilon > 0$, a comparison is conducted against the state-of-the-art method outlined by Silva et al.[28], as depicted in Fig. 7a. Proposition 2 yields a shallower circuit than the method[28] as soon as $n \gtrsim 10^5$.

Finally, with a fixed number of control qubits set at $n = 100$, Fig. 7b illustrates the depth as a function of the number of ancillae. For a single ancilla, the use of the circuit in Fig. 1 already surpasses the state-of-the-art, and an increase in the number of zeroed ancillae rapidly reduces the overall circuit depth. Proposition 3 achieves the same depth as the previous best method but with a significantly lower number of ancillae, getting even larger as the number $n$ of control qubits increases as depicted in Fig. 7b, c.

Table 1 compares the depth of various methods without employing any ancilla by fitting numerically the data from the implementations. Craig Gidney's method stands out as the only exact method but results in a notably higher overall depth. The linear decomposition from[28] achieves a leading coefficient of 48 at an approximation error of $\epsilon = 10^{-7}$, representing a significant improvement. Lastly, the implementation of Proposition 2 delivers a depth of $\lceil \log(\pi/\epsilon) \rceil (43 \log(n)^3 - 1287)$ and emerges as the most efficient method for $n \gtrsim 10^5$ control qubits.

In Table 2, various methods are compared when using a borrowed ancilla qubit. It is worth noting that employing such methods allows putting to use any qubit unaffected by the controlled operation for implementation, a situation frequently encountered in practice without the need for extra qubits. Notably, Proposition 1 is competitive across all control qubit numbers. The advantage becomes evident with over $10^2$ control qubits, resulting in a significant decrease in circuit depth. From this table, it is also straightforward to compare the depth of controlled-SU(2) gates. Reference[29] improved the base construction presented in ref. [15] to yield a depth of leading-order term $55n$. Recently[30], improved this result with a depth of $32n$. Naively, the construction from Corollary 2 has a a leading order depth of two times the depth of a $C^{n-1}(X)$ with borrowed ancilla qubit: $86 \log(n)^3$. By implementing the first $C^{n-1}(X)$ with the decomposition shown in Fig. 2 ($\mathfrak{C} \, \mathcal{C}_{R_0}^a \, \mathfrak{C} \, \mathcal{C}_{R_0}^a$) and the second with its conjugate ($\mathcal{C}_{R_0}^a \, \mathfrak{C} \, \mathcal{C}_{R_0}^a \, \mathfrak{C}$), this depth

**Table 3 | Circuit depth of zeroed ancillae methods**

| Method | Circuit depth | Zeroed |
|---|---|---|
| Barenco 1[15] | $36n - 111$ | 1 |
| Barenco $n - 2$[15] | $12n - 12$ | $n - 2$ |
| Proposition 1 | $27\log(n)^3 - 808$ | 1 |
| Proposition 3 | $27\log(n/\lceil m/2 \rceil)^3 +$ | $2 \leq m \leq n$ |
| | $16\lceil \log(\lfloor m/2 \rfloor) \rceil - 808$ | |
| He[22] | $16\lceil \log(n) \rceil + 12$ | $n$ |

The depths are numerically fitted in the range from $10^2$ to $10^7$ control qubits.

can be reduced by $2 \times 43\log(n^{1/2})^3$ bringing a leading order of $76\log(n)^3$.

Finally, Table 3 compares the methods using additional zeroed ancilla qubits. For a given number of available zeroed ancillae, Proposition 3 achieves shallower circuits. When the number of ancilla qubits matches the number of control qubits, the method proves as effective as the previously shallowest method by He et al.[22]. However, note that similar performances can be reached with significantly less zeroed ancillae (see Fig. 7b and c).

This paragraph provides an overview of some standard quantum algorithm oracles where multi-controlled operations act as both building blocks and complexity drivers and where the improved decomposition reported in this paper readily provides the corresponding speedup. Quantum search[31], quantum phase estimation[8], and Hamiltonian simulation[10] provide robust support for the claimed exponential quantum advantage. These algorithms can be seen as particular instances of the quantum singular value transformation (QSVT)[6], which allows the embedding of any Hamiltonian $H$ into an invariant subspace of the signal unitary, thereby enabling to compute a broad range of polynomials of $H$. The qubitisation[5] is the central technique to the framework of QSVT. When qubitising the Hamiltonian expressed as a linear combination $H = \sum_{k=1}^{s} \alpha_k U_k$, with each $U_k$ decomposable into a maximum of $C$ native gates, the process exhibits optimal query complexity for the two following oracles. The PREPARE oracle consists of a quantum state preparation step. Formally, it involves preparing the state $|\text{PREPARE}\rangle$ from a set of coefficients $\{\alpha_k\}_{k=1}^{s}$ such as :

$$\text{PREPARE}\,|0\rangle^{\otimes \log s} = \sum_{k=1}^{s} \sqrt{\frac{\alpha_k}{\lambda}}|k\rangle, \lambda = \sum_{k=1}^{s} \alpha_k \quad (10)$$

The CVO-QRAM algorithm[32] performs an efficient task by proceeding s layers of n-controlled operations, where n represents the number of control qubits and s represents the number of non-zero amplitude in the target state. The resulting circuit exhibits a depth of $\mathcal{O}(sn)$ assuming the usual linear decomposition of multi-controlled operations. The $C^n(X)$ gate decompositions provided in this paper readily improve the scaling to $\mathcal{O}(s\log(n)^3)$. The PREPARE operator finds application in a broader context beyond qubitisation, whenever there is a need to transfer classical data into the qubit register. The SELECT oracle consists of a block-diagonal operator and acts as follows :

$$\text{SELECT} \equiv \sum_{k=1}^{s} |k\rangle\langle k| \otimes U_k \quad (11)$$

It can be efficiently decomposed as s layers of log(s)-controlled-$U_k$ operations, yielding a depth in $\mathcal{O}(s\log(s)C)$ with usual linear-depth decomposition of $C^{\log(s)}(U)$ gates. The decompositions presented in this paper readily reduce the SELECT operation complexity to $\mathcal{O}(s\log(\log(s))^3 C)$.

Considering an example application in ground-state quantum chemistry, the quantum phase estimation (QPE) stands as the standard fault-tolerant algorithm. The QPE performs a projection on an eigenstate of the Hamiltonian and provides an estimate of the associated eigenenergy. The overall complexity is determined by the preparation of an accurate initial state, and the implementation of the phase estimation circuit. A possible strategy involves using CVO-QRAM to initialise the quantum register to an accurate approximation of the ground state obtained with classical computational chemistry simulations[33]. Then, the phase estimation algorithm calls multiple times a controlled unitary $U$. The unitary $U$ should be encoded so that its spectrum is related to the spectrum of the molecular Hamiltonian $H$. The qubitisation outlined above has become a standard for this task, for example, by implementing the quantum walk operator $U = e^{i\arccos H}$[6]. The polylogarithmic-depth $C^n(X)$ operations presented in this paper directly reduce the depth of two building blocks in QPE, leading to corresponding improvements in the algorithm. The expected costs associated with achieving a quantum advantage in chemistry with QPE[34–38] require a reconsideration incorporating the previous enhancements.

In summary, this paper introduced three methods for decomposing $C^n(X)$ gates into arbitrary single-qubit and CNOT gates. Proposition 1 takes advantage of a single borrowed ancilla qubit to implement a $C^n(X)$ gate with depth complexity $\Theta(\log(n)^3)$. Such polylogarithmic complexity significantly improves the present state-of-the-art which is set at a depth of $\Theta(n)$. Proposition 2 aims at the more general task of controlling arbitrary single-qubit unitaries. Introducing an approximation error $\epsilon > 0$ and making use of the previous proposition, the task can be achieved with depth $\mathcal{O}(\log(1/\epsilon)\log(n)^3)$. With polylogarithmic dependence on both relevant parameters, this approach emerges as the most efficient in its category. When zeroed ancillae are available, Proposition 3 provides a strategy for enhancing the efficiency of a $C^n(X)$ gate by optimising the utilisation of these ancillae. A glimpse into the potential applications of these enhanced $C^n(X)$ implementations within the QSVT framework is presented.

While Propositions 1 and 2 offer a superpolynomial speedup in terms of depth, the size is increased by a polylogarithmic factor. The size of the circuit, especially the count of non-Clifford gates (T gates, Toffoli,...), can dominate the total execution time due to the necessity of preparing magic states through distillation[39]. This process is highly resource-intensive in terms of both runtime and ancilla count, and it does not always allow for efficient parallel execution. Therefore, the larger size resulting from the proposed decompositions could be seen as a drawback compared to methods that scale linearly in size. The key aim of these decompositions is to pioneer a novel approach to circuit design that prioritises minimising depth.

It is also important to note that, for a large number $n$ of qubits ($n \geq 10^5$), it is likely that an error lower than $2^{-n}$ is experimentally hard to achieve on the parameters of the quantum gates, even in the context of fault-tolerant quantum computing with error correction. Therefore, depending on the wavefunctions that are manipulated in the QPU, it might be preferable to skip a large controlled operation (with more than $10^5$ controls). Examples, where gates with exponentially small phases are omitted can be seen in the Approximate Quantum Fourier Transform[40] or with Proposition 2. Conversely, removing a multi-controlled NOT gate could significantly impact the algorithm's outcome in the case of sparse quantum state preparation for example[33].

Further research could explore the potential benefits these enhanced multi-controlled NOT decompositions offer across various quantum algorithms. It could also aim to determine whether the new circuit design behind these decompositions can lead to other improved Oracle implementation strategies.

## Data availability

The data generated in this study have been deposited in the database under accession code[41].

## Code availability

The code generated in this study has been deposited in the following repository[41].

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

## Author contributions

B.C., C.F., and J.Z. conceived and designed the experiments. B.C., C.F., and J.Z. performed the experiments. B.C., C.F., J.Z., F.D., A.P., and J.P.P. analyzed the data. B.C., C.F., and J.Z. contributed materials/analysis tools. B.C., C.F., J.Z., A.P., and J.P.P. wrote the paper. F.D., A.P., and J.P.P. supervised the work.

## Competing interests

JPP is a shareholder and co-founder of Qubit Pharmaceuticals. The remaining authors declare no other competing interests.
