## [Peer Review File · Nature Communications]

Polylogarithmic-depth controlled-NOT gates without ancilla qubitsREVIEWER COMMENTS

Reviewer #1 (Remarks to the Author):

In this paper the authors describe three novel constructions of $C^n(X)$, decomposing it into CNOT and arbitrary single-qubit gates. These multi-controlled operations are quite common in quantum algorithms applications. Construction of these circuits is important and optimization of them is non-trivial. Here the authors achieve an exponential improvement in the circuit depth over previous constructions, for some cases. This do come at the cost of total number of gates. While this is a significant improvement of depth , following are some of the major drawbacks of this paper.

1. This paper is very poorly written. In fact, it becomes tough to follow the arguments at places. It required extra effort to be even convinced about the correctness of the constructions. Much of the confusions arise due to notations that are either poorly defined or have been defined or used in very confusing ways. Some of these I point out in the following points.
2. Page 3 first paragraph : $R_i = \{q_j\}...$ for each i beginning from 0 ? This should be 1.
3. Appendix A : When the operations \oplus , \boxplus between two registers are defined, it should be clear how the states of the registers change. One way of defining can be as follows. $\oplus: \ket{R_1}\ket{R_2} \rightarrow \ket{R_1'}\ket{R_2'}$. Else, it is difficult to follow.
4. Appendix A first paragraph : " $R \oplus R'$ is defined to be 1 if exactly one of the two registers is active..." - What does this mean? Where is this 1 stored ? This is crucial to understand \boxplus and other equations in this page.
5. Appendix A, line 6 : Control qubits are grouped into $b+1$ sub-registers.
6. Appendix A after Equation A3 : Equation A4 proves step by step...It seems Equation A4 is wrongly placed.

7. Appendix A first line in set of equations including A4 : How does 1-qubit 0 change to multi-qubit R_0 ? R_0 has $2p$ qubits. Again it is important to define \oplus properly.

8. The following two lines are also not clear because of poor definition of \boxplus .

9. Similar confusions arise in the set of equations at the beginning of page 11.

10. Section III : I have difficulty following the numerical results. For example, in previous sections asymptotic expressions of circuit depth have been provided. Then how do you justify the constants in Table I, II, III ? The authors should give a more detail explanations about how the experiments have been conducted.

11. From which results of [15] does the constants in Table I, II, III follow ?

12. Do all the previous works cited in Table I, II, III compute circuit depth ? Or, they compute circuit size, which becomes a trivial bound on circuit depth ? This information is important in order to have a better picture.

13. The circuit depth for zeroed ancilla method is much worse than the one in [22], which I think is one of the few cited papers that properly compute a bound on depth.

14. It seems the achieved reduction in circuit depth comes at the cost of circuit size. There is a multiplicative factor of polynomial in $\log n$. This can be a significant factor and the authors should highlight this for better perspective. Depending upon the applications this can even nullify the advantage of a reduced depth.

15. The authors have used "controlled-NOT" in many places instead of multi-controlled-NOT or n -controlled-NOT. This can confuse the reader between CNOT and $C^n(X)$.

In view of all the above points I do not think this paper meets the mark for publication in this journal.

Reviewer #2 (Remarks to the Author):

The manuscript "Polylogarithmic-depth controlled-NOT gates without ancilla qubits" provides a decomposition of multicontrolled not gates into controlled-NOT and single qubit gates with polylogarithmic depth using a divide-and-conquer approach. This is the main result of the manuscript, because previous decompositions of multicontrolled gates have a linear depth. The main results are described in Sections II.A.1 and II.A.2. The rest of the manuscript describes implications of the main results.

The result is important for implementation of quantum algorithms in quantum devices and the idea is sound and simple to understand from figures 1 and 2.

Multicontrolled quantum gates are a building block for development of quantum algorithms. The proposed method can impact all quantum algorithms that requires multicontrolled quantum gates, including traditional algorithms as quantum search and more recent algorithms as pointed in the manuscript.

The description of the algorithm is clear and sound. The authors compare the main result with previous methods in the literature. However, the secondary results as the decomposition of $SU(2)$ multicontrolled gates and the adjustable-depth method are not compared with previous methods.

The numerical analysis could include a comparison of the $SU(2)$ multicontrolled gate with Ref. 15, theorem 5 of "Iten, Raban, et al. "Quantum circuits for isometries." *Physical Review A* 93.3 (2016): 032318." and "Vale, Rafaella, et al. "Circuit decomposition of multi-controlled special unitary single-qubit gates." *IEEE Transactions on Computer-Aided Design of Integrated Circuits and Systems* (2023)." The adjustable depth method could be compared with the Ref. 22.

This manuscript presents an important advance in the decomposition of quantum operators that has the potential to impact the quantum computation community. I am in favour of acceptance of the article.

I did not assess fully Section IV and appendix.

Reviewer #3 (Remarks to the Author):

I think the paper is correct, and that it contains an interesting circuit decomposition idea. There are a two issues I think must be fixed for the paper to be publishable. Therefore I suggest revise-and-resubmit.

1. The paper needs some computer parsable files with examples of the full construction. This could be a python program that generates the circuit, or a Q# program implementing the circuit, or a collection of hardcoded .qasm files for a variety of problem sizes, or etc. Without such files, it's too difficult to really be sure that the construction performs correctly and uses the claimed number of gates.

2. I like improving circuits, but the paper needs to stop overselling the benefits of reducing depth. The abstract of the paper claims the work will create "exponential speedup of countless quantum algorithms". This is just not true. The issue is that large scale quantum algorithms don't bottleneck on circuit depth; they bottleneck on things like magic state distillation. Lowering the depth of the MCX gate can't speed up an algorithm if the algorithm was already spending its time waiting for enough Toffoli states to be made; lowering the depth will just make it wait for states more intensely. Actually, since these low depth circuits pay a polylogarithmic Toffoli count overhead to achieve the depth, they're likely to finish **slower** in practice. On the other end of the spectrum, in the asymptotic limit of arbitrarily many factories where logarithmic depth does become viable, you would repurpose some of the factories into clean ancilla qubits for use by the construction. The ideas in this paper are interesting because they extend our knowledge of circuit construction, not because they would be used in practice.

A potential issue I noticed, depending on how theoretical vs practical your sensibilities are, is that the figures plot out the cost of the circuit to pretty absurd sizes. For example, FIG7 extrapolates out to tens of millions of control qubits. The issue with projecting this far is that having ten million controls means the gate affects 1 out of every $2^{10000000}$ amplitudes. For the average wavefunction you'd expect to run into in a quantum algorithm, this means

the operation's effect is akin to rotating by $2^{-10000000}$ of a turn. Therefore you should simply skip the operation. In fact, skipping it would implement it to a higher fidelity than doing it because, even with error correction, operations are done to a tolerance far worse than $2^{-10000000}$ of a turn. (You would need to double-check that the superposition can't concentrate on the case where all the controls were on in the context the gate appears, but I suspect in almost all situations this would hold.) In practice, the way FIG7 would actually play out is that the cost would rise until somewhere between 100 controls and 10000 controls, then drop to zero as skipping became favorable. As an example of this kind of operation discarding actually being used, see <https://arxiv.org/abs/1803.04933>. That paper performs additions that, strictly speaking, must have length n (think 2048) to be correct. But they are truncated to a length $O(\log(n))$ (think ~ 100) to reduce costs, at the cost of a negligible error rate.

Referee 1

In this paper the authors describe three novel constructions of $C^n(X)$, decomposing it into CNOT and arbitrary single-qubit gates. These multi-controlled operations are quite common in quantum algorithms applications. Construction of these circuits is important and optimization of them is non-trivial. Here the authors achieve an exponential improvement in the circuit depth over previous constructions, for some cases. This do come at the cost of total number of gates. While this is a significant improvement of depth, following are some of the major drawbacks of this paper.

Response: We thank the reviewer for their time and effort. We believe the revised manuscript is significantly improved and fully addresses all the concerns raised.

Comment: *This paper is very poorly written. In fact, it becomes tough to follow the arguments at places. It required extra effort to be even convinced about the correctness of the constructions. Much of the confusions arise due to notations that are either poorly defined or have been defined or used in very confusing ways. Some of these I point out in the following points.*

Response: We thank the reviewer for pointing out this problem.

We largely re-wrote Appendix A, simplifying the notation, and emphasizing key technical points, which help following the calculation steps.

We added further explanations to the calculations of the numerical fits reported in Table I-III.

We have also added further discussions on the circuit size, explaining potential impact on practical implementation of our results.

We believe the revised version of the manuscript is now clear and understandable. We reply to each of the concerns below.

Comment: *Page 3 first paragraph : $R_i = \{q_j\}...$ for each i beginning from 0 ? This should be 1.*

Response: Thank you for pointing out this mistake. It has been corrected.

Comment: *Appendix A : When the operations \oplus , \boxplus between two registers are defined, it should be clear how the states of the registers change. One way of defining can be as follows. $\oplus : |R_1\rangle |R_2\rangle \rightarrow |R'_1\rangle |R'_2\rangle$. Else, it is difficult to follow.*

Response: We thank the reviewer for their comment.

We decided to redefine the \oplus and \boxplus symbols as follows:

The operator $\oplus : \{0, 1\}^2 \rightarrow \{0, 1\}$ is used to denote addition modulo 2, while $\boxplus : \cup_{n=1}^{\infty} \{0, 1\}^n \times \{0, 1\}^n \rightarrow \cup_{n=1}^{\infty} \{0, 1\}^n$ signifies bitwise addition modulo two: $(x_1, \dots, x_n) \boxplus (y_1, \dots, y_n) = (x_1 \oplus y_1, \dots, x_n \oplus y_n)$.

Comment: *Appendix A first paragraph : " $R \oplus R'$ is defined to be 1 if exactly one of the two registers is active..." - What does this mean? Where is this 1 stored ? This is crucial to understand \boxplus and other equations in this page.*

Response: Here we want a compact notation to compare the activation states of two registers, resulting in a 1 if and only if they are different. To improve the notation, we introduced new functions c and \bar{c} :

Let $c : \cup_{n=1}^{\infty} \{0, 1\}^n \rightarrow \{0, 1\}$, $(x_1, \dots, x_n) \mapsto \prod_{j=1}^n x_j$. Also, let $\bar{c} : \cup_{n=1}^{\infty} \{0, 1\}^n \rightarrow \{0, 1\}$, $(x_1, \dots, x_n) \mapsto \prod_{j=1}^n (1 - x_j)$.

What was previously written as $R \oplus R'$ is now written $c(R) \oplus c(R')$ and can indeed only take values in $\{0, 1\}$, justifying its use as a label for a single-qubit computational basis state.

Comment: *Appendix A, line 6 : Control qubits are grouped into $b+1$ sub-registers.*

Response: We thank the reviewer for pointing this out. The first paragraph of Appendix A has been significantly modified. Now the way the states are labelled is introduced before Lemma 2.

Comment: *Appendix A after Equation A3 : Equation A4 proves step by step...It seems Equation A4 is wrongly placed.*

Response: We believe with the updates this equation is now properly placed (now Equation A8).

Comment: *Appendix A first line in set of equations including A4 : How does 1-qubit R_0 change to multi-qubit R_0 ? R_0 has $2p$ qubits. Again it is important to define \boxplus properly.*

Response: As discussed in our response above, we now use the notation $c(R_0)$. The ancilla is in the state 1 if and only if $c(R_0) = 1$, meaning that the register R_0 is active.

Comment: *The following two lines are also not clear because of poor definition of \boxplus .*

Response: We agree with the reviewer, the notation was too compact to reflect the amount of information it contained.

We implemented the following changes. First, the computational basis states are now always written under the same form as $|R_0^*, R_0^b, R \setminus R_0, a, t\rangle$. R_0^* has b qubits, R_0^b has $2p - b$ qubits, $R \setminus R_0$ has $n - 2p$ qubits, a has a single qubit, and t has a single qubit. We always write the kets as $|\alpha, \beta, \gamma, \theta, \delta\rangle$ with α containing b qubits, β containing $2p - b$ qubits, γ containing $n - 2p$ qubits, θ containing a single qubit and δ containing a single qubit. At each line, the state of only one of these subregisters changes. We hope this change will make the steps easier to follow. Second, what was previously written $R_0^* \boxplus (R_1, \dots, R_b)$ is now written $R_0^* \boxplus (c(R_1), \dots, c(R_b))$.

In the revised Appendix 1, we have also added, after Lemma 1, an explanation to help understanding the notation \boxplus :

$$\text{It can be noted that } R_0^* \boxplus (c(R_1), \dots, c(R_b)) = (q_0 \oplus c(R_1), \dots, q_{b-1} \oplus R_b).$$

Comment: *Similar confusions arise in the set of equations at the beginning of page 11.*

Response: We believe this has been clarified in the response to the point above.

Comment: *Section III : I have difficulty following the numerical results. For example, in previous sections asymptotic expressions of circuit depth have been provided. Then how do you justify the constants in Table I, II, III ? The authors should give a more detail explanations about how the experiments have been conducted.*

Response: We thank the reviewer for pointing this. The 'Numerical Analysis' section now starts with a paragraph aiming at answering this concern:

The circuits are compiled in the basis of single-qubit and CNOT gates for number of control qubits ranging from 10^2 to 10^7 . The obtained depths are numerically fitted, giving useful estimates for which method to implement depending on the controlled operation to perform. Since this section only aims to serve as a resource estimator, the linear depth decompositions are fitted with a first-order polynomial in the variable n and the depth of the decomposition from Proposition 1 is fitted with a first-order polynomial in the variable $\log(n)^3$.

The 'Scaling' term has been replaced by 'Circuit depth' in the column names. The range of number of control qubits used to compute the fits is also recalled in the captions.

Comment: *From which results of [15] does the constants in Table I, II, III follow ?*

Response: As for the previous response, this question is addressed in the newly introduced paragraph at the beginning of the 'Numerical Analysis' section.

Comment: *Do all the previous works cited in Table I, II, III compute circuit depth ? Or, they compute circuit size, which becomes a trivial bound on circuit depth ? This information is important in order to have a better picture.*

Response: He et al. computed the depth of their decomposition. The other methods saw their depth computed numerically, as explained in a previous response.

Comment: *The circuit depth for zeroed ancilla method is much worse than the one in [22], which I think is one of the few cited papers that properly compute a bound on depth.*

Response: The single-zeroed-ancilla method indeed provides shallower circuits than the most efficient construction in [22]. However, the latter construction requires as many zeroed ancilla qubits as there are control qubits. Our Proposition 3 provides a convenient way to get to comparable performances while requiring significantly less ancillae.

Comment: *It seems the achieved reduction in circuit depth comes at the cost of circuit size. There is a multiplicative factor of polynomial in $\log n$. This can be a significant factor and the authors should highlight this for better perspective. Depending upon the applications this can even nullify the advantage of a reduced depth.*

Response: We have added a discussion of circuit size to the paper.

The analysis presented in this paper was done in the basis of single-qubit and CNOT gates. The obtained circuit depth reduction could be achieved at the cost of a polylogarithmic increase in circuit size. However, depending on the hardware at hand, this overhead in non-Clifford gates may cause a quantum processing unit to wait for sufficiently many magic states to be created. Therefore, the exponential advantage in gate run time may be attenuated. In other cases, if Toffoli operations are available natively [1], this problem does not arise.

Comment: *The authors have used "controlled-NOT" in many places instead of multi-controlled-NOT or n-controlled-NOT. This can confuse the reader between CNOT and $C^n(X)$.*

Response: We have improved the manuscript to use multi-controlled NOT consistently.

Referee 2

The manuscript "Polylogarithmic-depth controlled-NOT gates without ancilla qubits" provides a decomposition of multicontrolled not gates into controlled-NOT and single qubit gates with polylogarithmic depth using a divide-and-conquer approach. This is the main result of the manuscript, because previous decompositions of multicontrolled gates have a linear depth. The main results are described in Sections II.A.1 and II.A.2. The rest of the manuscript describes implications of the main results. The result is important for implementation of quantum algorithms in quantum devices and the idea is sound and simple to understand from figures 1 and 2. Multicontrolled quantum gates are a building block for development of quantum algorithms. The proposed method can impact all quantum algorithms that requires multicontrolled quantum gates, including traditional algorithms as quantum search and more recent algorithms as pointed in the manuscript.

The description of the algorithm is clear and sound. The authors compare the main result with previous methods in the literature. However, the secondary results as the decomposition of $SU(2)$ multicontrolled gates and the adjustable-depth method are not compared with previous methods.

Comment: *The numerical analysis could include a comparison of the $SU(2)$ multicontrolled gate with Ref. 15, theorem 5 of "Iten, Raban, et al. "Quantum circuits for isometries." *Physical Review A* 93.3 (2016): 032318." and "Vale, Rafaella, et al. "Circuit decomposition of multi-controlled special unitary single-qubit gates." *IEEE Transactions on Computer-Aided Design of Integrated Circuits and Systems* (2023)." The adjustable depth method could be compared with the Ref. 22.*

Response: We thank the reviewer for their suggestions. We have added a comparison with these references in the numerical analysis section of the manuscript.

It is also straightforward to compare the depth of controlled- $SU(2)$ gates. Reference [2] improved the base construction presented in [3] to yield a depth of leading-order term $55n$. Recently, [4] improved this result with a depth of $32n$. Naively, the construction from Corollary 2 has a leading order depth of two times the depth of a $C^{n-1}(X)$ with borrowed ancilla qubit: $86 \log(n)^3$. By implementing the first $C^{n-1}(X)$ with the decomposition shown in Figure 2 ($\mathfrak{C} C_{R_0}^a \mathfrak{C} C_{R_0}^a$) and the second with its conjugate ($C_{R_0}^a \mathfrak{C} C_{R_0}^a \mathfrak{C}$), this depth can be reduced by $2 \times 43 \log(n^{1/2})^3$ bringing a leading order of $76 \log(n)^3$.

This manuscript presents an important advance in the decomposition of quantum operators that has the potential to impact the quantum computation community. I am in favour of acceptance of the article. I did not assess fully Section IV and appendix.

Referee 3

Comment: *The paper needs some computer parsable files with examples of the full construction. This could be a python program that generates the circuit, or a Q# program implementing the circuit, or a collection of hardcoded .qasm files for a variety of problem sizes, or etc. Without such files, it's too difficult to really be sure that the construction performs correctly and uses the claimed number of gates.*

Response: We thank the reviewer for their suggestion. A python script is now available at https://github.com/BaptisteClaudon/Polylog_MCX-public.

Comment: *I like improving circuits, but the paper needs to stop overselling the benefits of reducing depth. The abstract of the paper claims the work will create "exponential speedup of countless quantum algorithms". This is just not true. The issue is that large scale quantum algorithms don't bottleneck on circuit depth; they bottleneck on things like magic state distillation. Lowering the depth of the MCX gate can't speed up an algorithm if the algorithm was already spending its time waiting for enough Toffoli states to be made; lowering the depth will just make it wait for states more intensely. Actually, since these low depth circuits pay a polylogarithmic Toffoli count overhead to achieve the depth, they're likely to finish *slower* in practice. On the other end of the spectrum, in the asymptotic limit of arbitrarily many factories where logarithmic depth does become viable, you would repurpose some of the factories into clean ancilla qubits for use by the construction. The ideas in this paper are interesting because they extend our knowledge of circuit construction, not because they would be used in practice.*

Response:

We thank the reviewer for their comment, with which we agree. We have added a detailed discussion of the subtlety of the likely speedup in the last section.

While Propositions 1 and 2 offer a superpolynomial speedup in terms of depth, the size is increased by a polylogarithmic factor. The size of the circuit, especially the count of non-Clifford gates (T gates, Toffoli, ...), can dominate the total execution time due to the necessity of preparing magic states through distillation [5]. This process is highly resource-intensive in terms of both runtime and ancilla count, and it does not always allow for efficient parallel execution. Therefore, the larger size resulting from the proposed decompositions could be seen as a drawback compared to methods that scale linearly in size. The key aim of these decompositions is to pioneer a novel approach to circuit design that prioritises minimising depth.

Comment: *A potential issue I noticed, depending on how theoretical vs practical your sensibilities are, is that the figures plot out the cost of the circuit to pretty absurd sizes. For example, FIG7 extrapolates out to tens of millions of control qubits. The issue with projecting this far is that having ten million controls means the gate affects 1 out of every $2^{10000000}$ amplitudes. For the average wavefunction you'd expect to run into in a quantum algorithm, this means the operation's effect is akin to rotating by $2^{-10000000}$ of a turn. Therefore you should simply skip the operation. In fact, skipping it would implement it to a higher fidelity than doing it because, even with error correction, operations are done to a tolerance far worse than $2^{-10000000}$ of a turn. (You would need to double-check that the superposition can't concentrate on the case where all the controls were on in the context the gate appears, but I suspect in almost all situations this would hold.) In practice, the way FIG7 would actually play out is that the cost would rise until somewhere between 100 controls and 10000 controls, then drop to zero as skipping became favorable. As an example of this kind of operation discarding actually being used, see <https://arxiv.org/abs/1803.04933>. That paper performs additions that, strictly speaking, must have length n (think 2048) to be correct. But they are truncated to a length $O(\log(n))$ (think 100) to reduce costs, at the cost of a negligible error rate.*

Response: We also thank the reviewer for the useful comment that significantly improves the relevance of the manuscript. We have added a detailed discussion of this point in the last section.

It is also important to note that, for a large number n of qubits ($n \geq 10^5$), it is likely that an error lower than 2^{-n} is experimentally hard to achieve on the parameters of the quantum gates, even

in the context of fault-tolerant quantum computing with error correction. Therefore, depending on the wavefunctions that are manipulated in the QPU, it might be preferable to skip a large controlled operation (with more than 10^5 controls). Examples where gates with exponentially small phases are omitted can be seen in the Approximate Quantum Fourier Transform [6] or with Proposition 2. Conversely, removing a multi-controlled NOT gate could significantly impact the algorithm's outcome, in the case of sparse quantum state preparation for example [7].

-
- [1] Yosep Kim, Alexis Morvan, Long B. Nguyen, Ravi K. Naik, Christian Jünger, Larry Chen, John Mark Kreikebaum, David I. Santiago, and Irfan Siddiqi. High-fidelity three-qubit itoffoli gate for fixed-frequency superconducting qubits. *Nature Physics*, 18(7):783–788, 2022.
 - [2] Raban Iten, Roger Colbeck, Ivan Kukuljan, Jonathan Home, and Matthias Christandl. Quantum circuits for isometries. *Phys. Rev. A*, 93:032318, Mar 2016.
 - [3] Adriano Barenco, Charles H. Bennett, Richard Cleve, David P. DiVincenzo, Norman Margolus, Peter Shor, Tycho Sleator, John A. Smolin, and Harald Weinfurter. Elementary gates for quantum computation. *Physical Review A*, 52(5):3457–3467, November 1995.
 - [4] Rafaella Vale, Thiago Melo D. Azevedo, Ismael C. S. Araújo, Israel F. Araujo, and Adenilton J. da Silva. Decomposition of multi-controlled special unitary single-qubit gates, 2023.
 - [5] Earl T. Campbell, Barbara M. Terhal, and Christophe Vuillot. Roads towards fault-tolerant universal quantum computation. *Nature*, 549(7671):172–179, September 2017.
 - [6] Yunseong Nam, Yuan Su, and Dmitri Maslov. Approximate quantum fourier transform with $o(n \log(n))$ t gates. *npj Quantum Information*, 6(1), March 2020.
 - [7] César Feniou, Olivier Adjoua, Baptiste Claudon, Julien Zylberman, Emmanuel Giner, and Jean-Philip Piquemal. Sparse quantum state preparation for strongly correlated systems. *The Journal of Physical Chemistry Letters*, 15:3197–3205, 2024. PMID: 38483286.

REVIEWER COMMENTS

Reviewer #2 (Remarks to the Author):

This manuscript significantly improves the state of the art in the decomposition of multi-controlled quantum gates. I did not follow the appendix; however, I can affirm that the results of the main text are correct.

Minor comments.

- I suggest replacing Theta tilde and O tilde by Theta and O notation and display the logarithmic factor.

- After Eq. 1 q is used to describe the target. T has the same meaning?

- D_n is an integer and $D(k)$ a function. Should the authors use a different letter for the function?

- update published arxiv references

[4] Maronese, Marco, et al. "Quantum compiling." Quantum Computing Environments. Cham: Springer International Publishing, 2022. 39-74.

[13] Peral-García, David, Juan Cruz-Benito, and Francisco José García-Peñalvo. "Systematic literature review: Quantum machine learning and its applications." Computer Science Review 51 (2024): 100619.

[21] Yuan, Pei, Jonathan Allcock, and Shengyu Zhang. "Does qubit connectivity impact quantum circuit complexity?." IEEE Transactions on Computer-Aided Design of Integrated Circuits and Systems (2023).

[25] Shende, Vivek V., and Igor L. Markov. "On the CNOT-cost of TOFFOLI gates." Quantum Information & Computation 9.5 (2009): 461-486.

[31] Vale, Raffaella, et al. "Circuit decomposition of multi-controlled special unitary single-

qubit gates." IEEE Transactions on Computer-Aided Design of Integrated Circuits and Systems (2023).

- If you are using bibtex, you can use {} to keep uppercase letters in the reference. For instance {T}offoli instead of Toffoli.

Reviewer #2 (Remarks on code availability):

The results of the paper are reproducible. I was able to run the code.

The README does not contain instructions for running the applications. README should point to the file paper_data.ipynb.

The CNOT count does not use qiskit standard method to count operations. Probably to avoid the transpilation cost. This could be explained in the github repo.

Reviewer #3 (Remarks to the Author):

I recommend revise-and-resubmit again.

I want to thank the authors for making code available at
https://github.com/BaptisteClaudon/Polylog_MCX-public

The code at this repository looks like it produces the construction as requested, but unfortunately I need to ask for a few improvements so that it can actually be verified.

Once the code is more verifiable, I'm happy to recommend publication.

REQUESTED IMPROVEMENT TO CODE

The basic problem is that the files in `initialise_log3/` decompose all the way down to quantum gates (like CX,H,T) instead of stopping at classical gates (Toffoli,CX,X). The authors are bootstrapping all their constructions from these hardcoded cases, which is reasonable for ensuring they have low gate counts, but a big problem for verifying the constructions. It makes simulation intractable.

For example, log3_3.qasm contains this:

```
OPENQASM 2.0;
include "qelib1.inc";
qreg q[4];
cx q[0],q[2];
u(pi/2,0,pi) q[3];
cx q[2],q[3];
u(0,0,-pi/4) q[3];
cx q[1],q[3];
u(0,0,pi/4) q[3];
cx q[2],q[3];
u(0,0,pi/4) q[2];
u(0,0,-pi/4) q[3];
cx q[1],q[3];
cx q[1],q[2];
u(0,0,pi/4) q[1];
u(0,0,-pi/4) q[2];
cx q[1],q[2];
cx q[0],q[2];
u(0,1.4065829705916304,-0.6211848071941821) q[3];
cx q[2],q[3];
u(0,0,-pi/4) q[3];
cx q[1],q[3];
u(0,0,pi/4) q[3];
cx q[2],q[3];
```

```
u(0,0,pi/4) q[2];
u(0,0,-pi/4) q[3];
cx q[1],q[3];
cx q[1],q[2];
u(0,0,pi/4) q[1];
u(0,0,-pi/4) q[2];
cx q[1],q[2];
u(pi/2,0,-3*pi/4) q[3];
```

instead of this:

```
OPENQASM 2.0;
include "qelib1.inc";
qreg q[4];
ccx q[0],q[1],q[2]
```

(As a side note, this particular file strikes me as being strangely inefficient. It contains 14 T gates, but a CCX gate should only be using 4 T gates when a clean ancilla is available (see <https://arxiv.org/abs/1212.5069>).

I recommend replacing these hardcoded files, or making a second set of files, that terminates at classical gates (Toffoli, CX, X) instead of quantum gates. Then I could sample huge circuits, like the $n=10000$ circuit, by producing n bits of my choosing, running them through the circuit using a classical simulator, and seeing if the bits changed in the correct way.

In principle, I could first verify the 30 hardcoded cases were actually MCX2 through MCX31 using a state vector simulator, and then replace them with just a classical gate, and then run the replaced-with-classical-gate code to get simulable constructions to verify. But I think this is more work than should be necessary for a third party to verify that the code works.

To be very concrete, what the code needs is this method:

```
...  
  
def mcx_flat_construction(n: int) -> qiskit.Circuit:  
    """Returns the construction as a qiskit circuit where the qiskit circuit  
    contains *only* X, CX, and CCX gates."""  
    ...  
    ...
```

Alternatively, this method:

```
...  
  
def mcx_toffolis(n: int) -> list[int | tuple[int, int] | tuple[int, int, int]]:  
    """Builds the n-qubit construction, and returns its X, CX, and Toffoli  
    gates. Each int in the resulting list is an X gate, each (int, int) is  
    a CX gate where the second int is the target qubit index. Each  
    (int, int, int) is a Toffoli gate where the third int is the target  
    qubit index."""  
    ...  
    ...
```

Or this method:

```
...  
  
def apply_construction_to(bits: list[bool]) -> list[bool]:  
    """Builds the n=len(bits) construction, simulates applying it to the  
    given bits, and returns the resulting output bits. Runs in less than  
    10 seconds when given a list of 10000 bits."""  
    ...  
    ...
```

The repository should also include at least one test, checking the construction

produces the correct result in at least one case. For example, assuming you added the `apply_construction_to` method mentioned above, you could then create a file `construction_test.py` containing this code:

```
'''
def test_1000_qubit_case_works_when_nearly_all_bits_are_on():
    for off_qubit in range(1000):
        in_bits = [True] * 1000
        in_bits[off_qubit] = False

        expected_out_bits = list(in_bits)
        if off_qubit == 999:
            expected_out_bits[999] ^= True

        out_bits = apply_construction_to(in_bits)
        assert out_bits == expected_out_bits
'''
```

Installing `pytest` and running `pytest construction_test.py` would then run this method and confirm it didn't raise an assertion.

CODE NITPICKS

The authors don't need to fix these things in order for me to recommend publication. But they would be good things to fix.

The repository contains several unnecessary files that could be removed (using `git rm`) and then ignored by adding a `.gitignore` file with the following contents:

```
.idea
```

.DS_Store
.ipynb_checkpoints
__pycache__

It's customary to include a `requirements.txt` file listing the packages required for the code to function. In this case the file would just contain:

```
qiskit
```

The README.md file should include a link to the tutorial notebook, to raise its visibility.

The tutorial notebook should end with printing out a full diagram of a construction, rather than an opaque

```
"<qiskit.circuit.instructionset.InstructionSet at 0x11c370c70>".
```

Referee 2

Comment: *This manuscript significantly improves the state of the art in the decomposition of multi-controlled quantum gates. I did not follow the appendix; however, I can affirm that the results of the main text are correct.*

Minor comments.

I suggest replacing Theta tilde and O tilde by Theta and O notation and display the logarithmic factor.

Response: Thanks for the suggestion. We gave up the use of \tilde{O} and $\tilde{\Theta}$.

Comment: *After Eq. 1 q is used to describe the target. T has the same meaning?*

Response: The notation q refers a white control qubit, for the X gate to be applied to the target qubit T . We have clarified this in the following sentence:

...where X_q denotes an X gate on white control qubit q .

Comment: *D_n is an integer and $D(k)$ a function. Should the authors use a different letter for the function?*

Response: We have replaced $D(k)$ with $\mathcal{D}(k)$, to try and emphasize that D_n and $\mathcal{D}(k)$ are closely related while being able to differentiate their notations.

Comment: *update published arxiv references*

[4] Maronese, Marco, et al. "Quantum compiling." *Quantum Computing Environments*. Cham: Springer International Publishing, 2022. 39-74.

[13] Peral-García, David, Juan Cruz-Benito, and Francisco José García-Peñalvo. "Systematic literature review: Quantum machine learning and its applications." *Computer Science Review* 51 (2024): 100619.

[21] Yuan, Pei, Jonathan Allcock, and Shengyu Zhang. "Does qubit connectivity impact quantum circuit complexity?." *IEEE Transactions on Computer-Aided Design of Integrated Circuits and Systems* (2023).

[25] Shende, Vivek V., and Igor L. Markov. "On the CNOT-cost of TOFFOLI gates." *Quantum Information & Computation* 9.5 (2009): 461-486.

[31] Vale, Rafaella, et al. "Circuit decomposition of multi-controlled special unitary single-qubit gates." *IEEE Transactions on Computer-Aided Design of Integrated Circuits and Systems* (2023).

Response: Thanks for the suggestions. We have implemented the modifications.

Comment: *If you are using bibtex, you can use {} to keep uppercase letters in the reference. For instance {T}offoli instead of Toffoli.*

Response: Capital letters now appear in the biography.

Comment: *Remarks on code availability:*

The results of the paper are reproducible. I was able to run the code.

The README does not contain instructions for running the applications. README should point to the file paper_data.ipynb.

The CNOT count does not use qiskit standard method to count operations. Probably to avoid the transpilation cost. This could be explained in the github repo.

Response: We thank the reviewer for the comments. README now refers to the Jupyter notebooks as well as the test file.

We have explained in the github repo that regarding the computation of circuit depths, it is indeed much faster to compute circuits depths through dynamic programming during circuit compilation than to use the Qiskit method afterwards.

Referee 3

Comment: *I recommend revise-and-resubmit again. I want to thank the authors for making code available at https://github.com/BaptisteClaudon/Polylog_MCX-public. The code at this repository looks like it produces the construction as requested, but unfortunately I need to ask for a few improvements so that it can actually be verified. Once the code is more verifiable, I'm happy to recommend publication.*

Comment: *# REQUESTED IMPROVEMENT TO CODE The basic problem is that the files in 'initialise_log3' decompose all the way down to quantum gates (like CX,H,T) instead of stopping at classical gates (Toffoli,CX,X). The authors are bootstrapping all their constructions from these hardcoded cases, which is reasonable for ensuring they have low gate counts, but a big problem for verifying the constructions. It makes simulation intractable. For example, log3_3.qasm contains this: [...] instead of this [...]: (As a side note, this particular file strikes me as being strangely inefficient. It contains 14 T gates, but a CCX gate should only be using 4 T gates when a clean ancilla is available (see <https://arxiv.org/abs/1212.5069>). I recommend replacing these hardcoded files, or making a second set of files, that terminates at classical gates (Toffoli, CX, X) instead of quantum gates.*

Response: We thank the reviewer for suggesting to compile the novel circuits in the basis of X , $CNOT$ and Toffoli gates. This basis set makes the construction much easier to understand. We added a file `log3_mcX_x_cx_ccx.py` implementing this feature. It does so without relying on brute-force-optimised small controlled-NOT gates. Instead, it only relies on the manually implemented $C^1(X)$, $C^2(X)$, $C^4(X)$. Thus, all the produced circuits can be compiled in the desired basis set.

Comment:

Then I could sample huge circuits, like the $n=10000$ circuit, by producing n bits of my choosing, running them through the circuit using a classical simulator, and seeing if the bits changed in the correct way. In principle, I could first verify the 30 hardcoded cases were actually MCX2 through MCX31 using a state vector simulator, and then replace them with just a classical gate, and then run the replaced-with-classical-gate code to get simulable constructions to verify. But I think this is more work than should be necessary for a third party to verify that the code works. To be very concrete, what the code needs is this method:

```
'''
def mcx_flat_construction(n: int) -> qiskit.Circuit:
    """Returns the construction as a qiskit circuit where the qiskit circuit
    contains *only* X, CX, and CCX gates."""
    ...
'''
```

Alternatively, this method:

```
'''
def mcx_toffolis(n: int) -> list[int | tuple[int, int] | tuple[int, int, int]]:
    """Builds the n-qubit construction, and returns its X, CX, and Toffoli
    gates. Each int in the resulting list is an X gate, each (int, int) is
    a CX gate where the second int is the target qubit index. Each
    (int, int, int) is a Toffoli gate where the third int is the target
    qubit index."""
    ...
'''
```

Or this method:

```
'''
def apply_construction_to(bits: list[bool]) -> list[bool]:
    """Builds the n=len(bits) construction, simulates applying it to the
    given bits, and returns the resulting output bits. Runs in less than
```

```
10 seconds when given a list of 10000 bits."""
...
'''
```

The repository should also include at least one test, checking the construction produces the correct result in at least one case. For example, assuming you added the 'apply_construction_to' method mentioned above, you could then create a file 'construction_test.py' containing this code:

```
'''
def test_1000_qubit_case_works_when_nearly_all_bits_are_on():
    for off_qubit in range(1000):
        in_bits = [True] * 1000
        in_bits[off_qubit] = False
        expected_out_bits = list(in_bits)
        if off_qubit == 999:
            expected_out_bits[999] ^= True
        out_bits = apply_construction_to(in_bits)
        assert out_bits == expected_out_bits
'''
```

Installing 'pytest' and running 'pytest construction_test.py' would then run this method and confirm it didn't raise an assertion.

Response: We thank the reviewer for giving us such a clear request. We added a file construction_test.py. This file contains three test functions. The first one generates the unitary matrix corresponding to the controlled-NOT circuits with up to 10 control qubits. It then verifies that it corresponds with the unitary matrix generated by the Qiskit library for the same gate (.mcx function). A second test function generates a gate with a 100 control qubits. It takes a random computational basis state and checks that the gate acts as required. The last test function checks that, if all the control qubits are in state 1, the target state is flipped. To perform the verification, we used the Qiskit backend

```
Aer.get_backend('aer_simulator').
```

Unfortunately, it fails to run the verification with 1000 control qubits. However, we believe that our tests are convincing. Moreover, Appendix 1 has been significantly improved and should now convince the reviewer that the construction is correct.

Comment: # CODE NITPICKS The authors don't need to fix these things in order for me to recommend publication. But they would be good things to fix. The repository contains several unnecessary files that could be removed (using 'git rm') and then ignored by adding a '.gitignore' file with the following contents:

```
.idea
.DS_Store
.ipynb_checkpoints
__pycache__
```

It's customary to include a 'requirements.txt' file listing the packages required for the code to function. In this case the file would just contain: qiskit The README.md file should include a link to the tutorial notebook, to raise its visibility. The tutorial notebook should end with printing out a full diagram of a construction, rather than an opaque

```
"<qiskit.circuit.instructionset.InstructionSet at 0x11c370c70>"
```

Response: We thank the reviewer for the suggestion. We added a .gitignore file, the requirements.txt file and updated the README.md file. The tutorial now displays a circuit in the basis set of X , $CNOT$, and $C^2(X)$ gates. The code should now be significantly easier to understand.

REVIEWERS' COMMENTS

Reviewer #3 (Remarks to the Author):

The authors provided code I was able to verify, and therefore I suggest publication.

It's pretty funny to try to test a 100 qubit case using a state vector simulator. I'm not surprised it didn't finish. (I'm pretty surprised it didn't immediately crash with an out of memory error.)

Anyways, the circuits were now simple enough in structure that I could just write my own tests to verify the constructions were computing CCC..CX. Here is the passing test code:

```
...
```

```
import collections
```

```
import itertools
```

```
import random
```

```
import numpy as np
```

```
import pytest
```

```
import qiskit
```

```
from qiskit.compiler import transpile
```

```
from log_3_mcx_x_cx_ccx import log3_cnx
```

```
def bulk_simulate_result_of_applying_classical_circuit_to(circuit: qiskit.QuantumCircuit,
```

```
input_states: np.ndarray) -> np.ndarray:
```

```
    states = np.copy(input_states)
```

```
    buffer = np.zeros_like(states[0])
```

```
    for instruction in circuit:
```

```
        if instruction.operation.name == 'x':
```

```
            assert len(instruction.qubits) == 1
```

```

q = instruction.qubits[0]._index
np.bitwise_not(states[q], out=states[q])
elif instruction.operation.name == 'cx':
assert len(instruction.qubits) == 2
c = instruction.qubits[0]._index
t = instruction.qubits[1]._index
states[t] ^= states[c]
elif instruction.operation.name == 'ccx':
assert len(instruction.qubits) == 3
a = instruction.qubits[0]._index
b = instruction.qubits[1]._index
t = instruction.qubits[2]._index
np.bitwise_and(states[a], states[b], out=buffer)
states[t] ^= buffer
elif instruction.operation.name == 'barrier':
pass
elif instruction.operation.name == 'measure':
pass
else:
raise NotImplementedError(f'{instruction=}')
return states

```

```

def compute_depth(circuit: qiskit.QuantumCircuit) -> int:
qubit_depth = collections.defaultdict(int)
known = ['x', 'cx', 'ccx']
for instruction in circuit:
if instruction.operation.name in known:
qs = [q._index for q in instruction.qubits]
layer = max(qubit_depth[q] for q in qs) + 1
for q in qs:
qubit_depth[q] = layer

```

```
elif instruction.operation.name == 'barrier':
    pass
elif instruction.operation.name == 'measure':
    pass
else:
    raise NotImplementedError(f'{instruction=}')
return max(qubit_depth.values())
```

```
@pytest.mark.parametrize('num_controls', [10, 100, 200, 500, 1000])
```

```
def test_depth(num_controls: int):
    num_qubits = num_controls + 2
    gate = log3_cnx(ncontrol=num_controls)
    circuit = qiskit.QuantumCircuit(num_qubits)
    circuit.append(gate, list(range(num_qubits)))
    circuit = transpile(circuit, basis_gates=['x', 'cx', 'ccx'])
    depth = compute_depth(circuit)
    lg_n = num_controls.bit_length()
    expected = 27 * lg_n**3 - 808
    assert depth <= expected
```

```
@pytest.mark.parametrize('num_controls', [1, 2, 3, 4, 5, 6, 7, 8, 9, 10, 32, 64, 100])
```

```
def test_fuzz_random_cases(num_controls: int):
    num_samples = 1024
    num_qubits = num_controls + 2
    target_index = num_controls + 1

    gate = log3_cnx(ncontrol=num_controls)
    circuit = qiskit.QuantumCircuit(num_qubits)
    circuit.append(gate, list(range(num_qubits)))
    circuit = transpile(circuit, basis_gates=['x', 'cx', 'ccx'])
```

```
input_states = np.random.randint(  
low=0,  
high=(1 << 64) - 1,  
size=(num_qubits, num_samples // 64),  
dtype=np.uint64,  
)
```

```
controls_satisfied = ~np.zeros_like(input_states[0])
```

```
for k in range(num_controls):
```

```
controls_satisfied &= input_states[k]
```

```
expected = np.copy(input_states)
```

```
expected[target_index] ^= controls_satisfied
```

```
actual = bulk_simulate_result_of_applying_classical_circuit_to(circuit, input_states)
```

```
assert np.array_equal(actual, expected)
```

```
@pytest.mark.parametrize('num_controls', [1, 2, 3, 4, 5, 6, 7, 8, 9, 10, 32, 64, 100])
```

```
@pytest.mark.parametrize('num_off_controls', [0, 1, 2])
```

```
def test_low_weight_cases(num_controls: int, num_off_controls):
```

```
num_qubits = num_controls + 2
```

```
target_index = num_controls + 1
```

```
ancilla_index = num_controls
```

```
gate = log3_cnx(ncontrol=num_controls)
```

```
circuit = qiskit.QuantumCircuit(num_qubits)
```

```
circuit.append(gate, list(range(num_qubits)))
```

```
circuit = transpile(circuit, basis_gates=['x', 'cx', 'ccx'])
```

```
cases = list(itertools.combinations(range(num_controls), num_off_controls))
```

```

input_states = np.ones(
shape=(num_qubits, len(cases)),
dtype=np.bool_,
)
for shot_index, on_bits in enumerate(cases):
for q in on_bits:
input_states[q, shot_index] = False
input_states[ancilla_index, shot_index] = random.random() < 0.5
input_states[target_index, shot_index] = random.random() < 0.5
input_states = np.packbits(input_states, axis=1)

controls_satisfied = ~np.zeros_like(input_states[0])
for q in range(num_controls):
controls_satisfied &= input_states[q]
expected = np.copy(input_states)
expected[target_index] ^= controls_satisfied

actual = bulk_simulate_result_of_applying_classical_circuit_to(circuit, input_states)

assert np.array_equal(actual, expected)
...

```

Reviewer #3 (Remarks on code availability):

Tests would never have finished, which is silly. Using a classical simulator lets tens of thousands of cases be tested per second. Otherwise good.

Referee 3

Comment: *(Remarks to the Author): The authors provided code I was able to verify, and therefore I suggest publication. It's pretty funny to try to test a 100 qubit case using a state vector simulator. I'm not surprised it didn't finish. (I'm pretty surprised it didn't immediately crash with an out of memory error.) Anyways, the circuits were now simple enough in structure that I could just write my own tests to verify the constructions were computing CCC..CX. Here is the passing test code:*

[code]

(Remarks on code availability): Tests would never have finished, which is silly. Using a classical simulator lets tens of thousands of cases be tested per second. Otherwise good.

Response: We thank the referee for providing such a useful test file. Our previous file could verify that our gates performed the expected operation for up to 150 control qubits. This new file, now included in the Github repository, allows the verification to be carried with over 1000 control qubits. Additionally, it verifies the depth upper bound of $27 \log(n)^3 - 808$.